# Atomic Data Needs in Astrophysics: The Galactic Center "Scandium Mystery"

**Brian Thorsbro**

Department of Astronomy and Theoretical Physics, Lund University, Box 43, 221 00 Lund, Sweden; thorsbro@astro.lu.se

**Abstract:** Investigating the Galactic center offers unique insights into the buildup and history of our Galaxy and is a stepping stone to understand galaxies in a larger context. It is reasonable to expect that the stars found in the Galactic center might have a different composition compared to stars found in the local neighborhood around the Sun. It is therefore quite exciting when recently there were reports of unusual neutral scandium, yttrium, and vanadium abundances found in the Galactic center stars, compared to local neighborhood stars. To explain the scandium abundances in the Galactic center, we turn to recent laboratory measurements and theoretical calculations done on the atomic oscillator strengths of neutral scandium lines in the near infrared. We combine these with measurements of the hyper fine splitting of neutral scandium. We show how these results can be used to explain the reported unusual scandium abundances and conclude that in this respect, the environment of the Galactic center is not that different from the environment in the local neighborhood around the sun.

**Keywords:** Galactic center; stellar abundances; scandium; hyper fine splitting

## 1. Introduction

Understanding the formation and evolution of galaxies is one of the important questions in astrophysics today [1]. The Milky Way, our own Galaxy, is central as it is the galaxy we can observe in the greatest detail, both because we want to explore the world we live in and because we can use it to understand galaxies at large, assuming the mediocrity principle, i.e., that we are not special.

Our Galaxy is currently seen as a barred spiral galaxy with multiple stellar populations, often roughly classified as the halo, the thin and thick disks, the bulge, and the Galactic center [1].

Key components to understanding the formation and evolution of the Galaxy are understanding the star formation rate and the mass distribution of the formed stars [2]. The constituents of the Galaxy that are easiest to observe are its stars, and the Galactic evolution can be decoded from its stars by understanding stellar structure and evolution. Fortunately, stellar structure and evolution theories are well developed, and they allow us to model how stars evolve and eventually perish [3]. In particular, we can model what kind of chemical species are synthesized in stars and consequently distributed to the surrounding environment, from which new stars are born.

Combining galactic and stellar evolution models thus demands the study of chemical evolution models that predict the abundances of chemical species in stars at different times and locations [4]. The chemical composition of the photosphere of a star, which is the region of the star from which light escapes, changes during the life of the star as heavier elements settle. However, as the star turns into a red giant towards the end of its life, convection inside the star remixes the material, undoing the settling of the heavy elements. A few light element species produced by the fusion processes in the center of the star are transported up to the photosphere with this convection as well. Apart from that, however, one can approximately assume to observe the composition of the environment in which the star was born by observing the composition of its current photosphere. These observations are done

with stellar spectroscopy, and the abundance of chemical species in the photosphere is determined using abundance analysis, which thus enables comparisons between models and observations.

In our study, we focus on the Galactic center, which is interesting as it is an environment that is unique to the Galaxy, particularly because of the presence of the super massive black hole. This could lead to a different evolution of stars compared to the environment in the vicinity of our Sun. In order to be able to observe stars in the Galactic center, one has to turn to very bright giant stars and observe them in the infrared wavelength regions. More energetic light is absorbed and scattered away by the dust lying between the center and the Sun, and thus, it is not possible to observe the Galactic center in the visible light wavelength range.

One chemical species that is of concern here is neutral scandium. In early 2018, it was reported that unusually high amounts of scandium, together with yttrium and vanadium, might be present in the Galactic center stars [5]. A unique scandium abundance in the Galactic center would suggest that the center is a site for a new channel of nucleosynthesis of neutral scandium and possibly other elements. Such a trend is important to understand, especially when observing the centers of far away galaxies, which would be a natural choice to observe first, as the center is the most luminous part of a galaxy.

In this work, we summarize our findings first reported in Thorsbro et al. [6] and discuss the atomic data needs in astrophysics on the basis of this.

## 2. Results

### 2.1. Observations

We observed 18 stars in the Galactic center and eight stars located in the solar neighborhood. The exact information about the observations of these stars, as well as how they were analyzed were detailed by Thorsbro et al. [6], Ryde et al. [7], Rich et al. [8]. All of the stars were of similar stellar classification, denominated M giants, which means that they had an effective temperature[1] between 3000 and 4000 K. The solar neighborhood stars were observed as a control group to compare the Galactic center stars. The stars were observed with the Keck II telescope (10 m class telescope), mounted with the NIRSPECspectrograph [9]. The spectra of the stars were recorded in the near infrared wavelength region around 2 microns ($\sim$5000 cm$^{-1}$) with a resolving power R $= \frac{\lambda}{\Delta\lambda} \simeq$ 23,000.

In the spring of 2018, Do et al. [5] reported to have found evidence for unusual high scandium abundances in stars in the Galactic center located within a few parsecs from the super massive black hole of our Galaxy. Their observed stars displayed unusually strong scandium line features in their spectra. They argued that modeling the observed spectra with synthetic spectra from first principles was impossible unless one assumed an unusually high scandium abundance. They further compared their observed stars to a star found in a globular cluster, which was a high stellar density environment, and showed that the comparison star did not show the same unusually high abundances. It is worth noting that the comparison star had an effective temperature that was about 800 to 1000 K warmer compared to the their observed Galactic center stars.

In Figure 1, the spectra from our observations are shown as we published them in Thorsbro et al. [6]. Here, it is worth noticing that the scandium line features were strong for stars located in both the Galactic center and in the solar neighborhood. Further, for both environments, the strength of the scandium line features diminished as the temperature increased. We could thus immediately conclude that the environment of the star did not seem to be connected to the strength of the line features. To improve our understanding, we therefore investigated if there could be assumptions in the modeling from first principles that needed to be revisited.

---

[1]   effective temperature is defined by the corresponding black-body radiation curve with an equal amount of radiation to the radiation coming from the star.

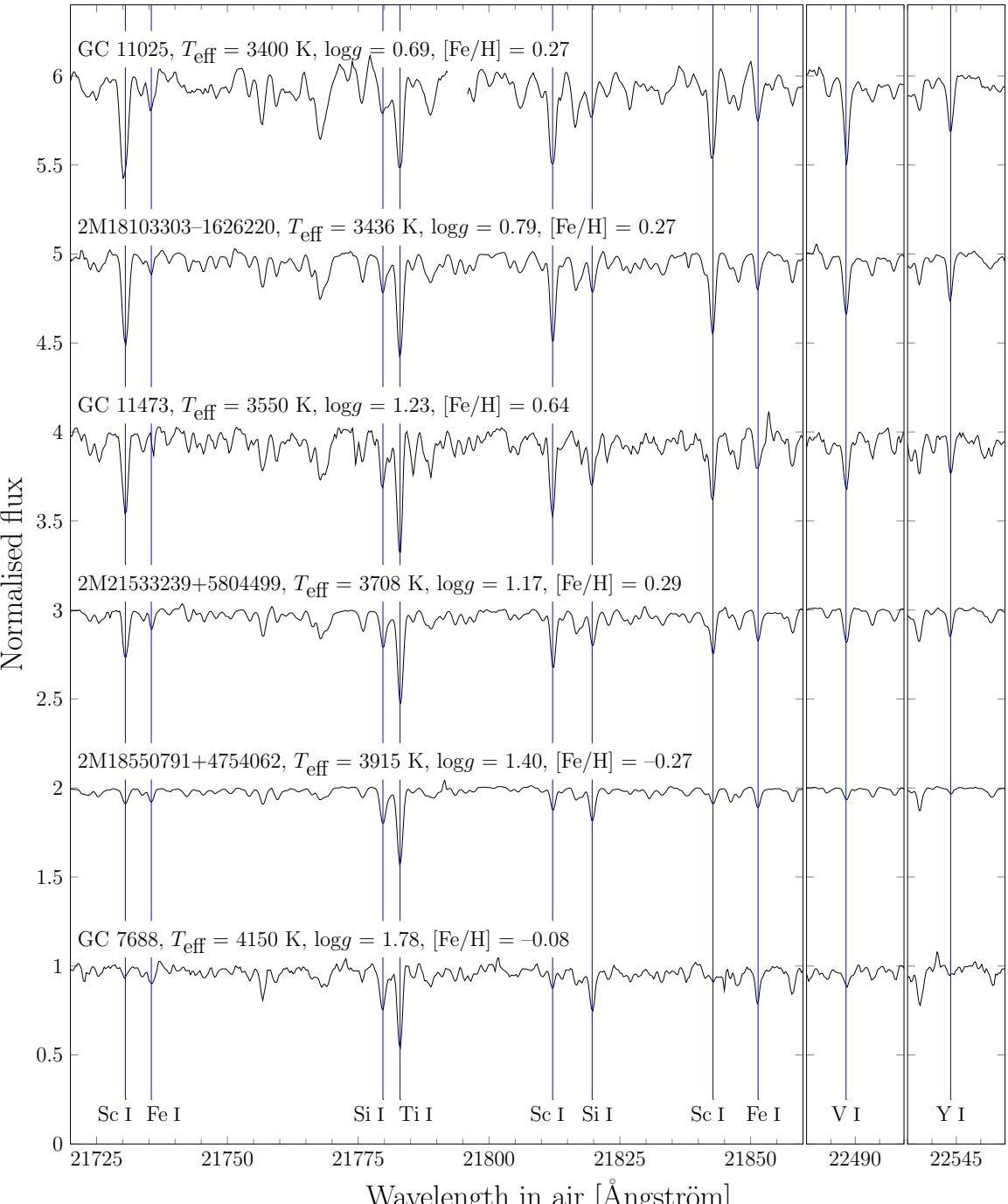

**Figure 1.** This figure from Thorsbro et al. [6] (their Figure 1) shows that the strong scandium feature seems to be connected to the effective temperature of the star. Six stellar spectra are plotted together with increasing temperature from top to bottom with blue vertical lines identifying lines of interest. The normalized fluxes are translated upward with integer values for presentation. The stars are a mix of three Galactic center stars (GC 7688, GC 11025, and GC 11473) and three solar neighborhood stars (2M18103303–1626220, 2M18550791+4754062, and 2M21533239+5804499). The spectra show strikingly strong scandium, vanadium, and yttrium lines in the cooler stars, even though the stars are located in widely different environments. As temperatures increase to 3900 K and beyond, the neutral lines of scandium, vanadium, and yttrium begin to vanish, presumably due to ionization.

## 2.2. Analyzing the Hyperfine Structure

The line transition in neutral scandium that is shown in Figure 1 is the $3d^24s$—$3d4s4p$ transition. Neutral scandium has a nuclear spin of $I = 7/2$, and since this transition involves an unpaired 4s-election, the hyperfine splitting is expected to be strong.

In general, there is a lack of experimental atomic data for near-IR transitions [10]. In response to this need, a program was initiated to provide accurate and vetted near-IR atomic data for stellar spectroscopy. This program was lead by Henrik Hartman and Per Jönsson at Malmö University in Sweden using the Edlén laboratory, a joint effort of Malmö and Lund universities.

Recent works measured the oscillator strengths and hyperfine structure of neutral scandium [11,12]. Both of these works utilized the Fourier transform spectrometer in the Edlén laboratory examining neutral scandium energized in a hollow cathode lamp.

In the work of Pehlivan et al. [11] intensity calibrated spectra of Sc I are used to experimentally determine branching fractions, which are then combined with radiative lifetimes from the literature [13–15] to derive accurate oscillator strengths for Sc I.

In the work of van Deelen [12], synthetic model spectra of Sc I hyperfine structure (HFS) multiplets were constructed and fitted to experimentally measured spectra. The results were compared to similar results from the literature [16–23], and investigations were initiated to examine the significant differences more closely. The work of van Deelen [12] can be considered the most recent and accurate compilation of Sc I HFS data for the near infrared wavelength region.

In the work of Thorsbro et al. [6], the theoretical line formation of a spectral Sc I line was modeled to explore the effects of both temperature and HFS, using the BSYNand EQWIDTHcodes based on routines from the MARCSmodel atmosphere code [24]. One-dimensional (1D) MARCS atmosphere models were used. These models were hydrostatic model photospheres in spherical geometry, computed assuming local thermodynamic equilibrium (LTE), chemical equilibrium, homogeneity, and conservation of the total flux (radiative plus convective, the convective flux being computed using the mixing length recipe [25]). The resulting line strength measured in equivalent width is plotted against temperature in Figure 2. For the spectral line based on a single atomic level transition, we used the measured oscillator strength from Pehlivan et al. [11]. For the HFS based spectral line, we used the combined work of Pehlivan et al. [11], van Deelen [12].

From Figure 2, it is clear that for the cooler stars, it was crucial to have the correctly modeled HFS to explain the stronger Sc I lines. That a Sc I line has an HFS means that in reality, it consists of many weaker lines. That the many weak lines appear as a single line is due to the fact that the resolution of the spectrometer is not high enough to resolve all the details and also due to different line broadening effects caused by temperature, pressure, and other conditions in the observed star that make the weak lines blend together. The onset of saturation is delayed as the many weaker lines individually saturate later compared to that of a singular strong line.

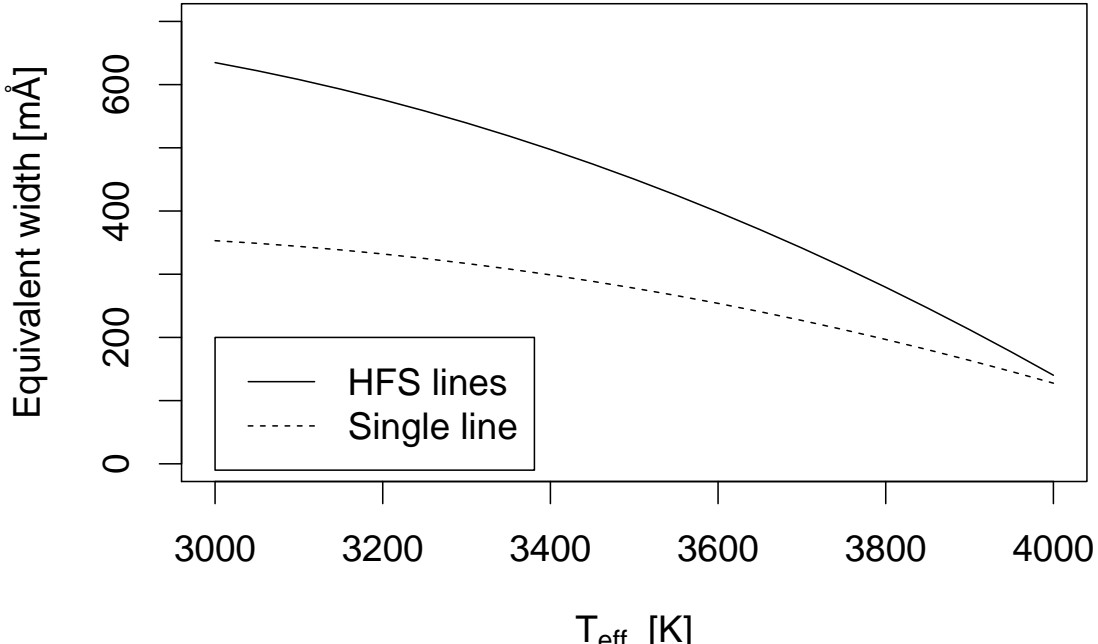

**Figure 2.** Figure from Thorsbro et al. [6] (their Figure 3). The equivalent width of a Sc I line as a function of temperature depends on whether the Sc I line is considered to be a combination of many small lines due to the hyperfine structure (HFS) or if the line is considered to be a singular atomic level transition. Notice how at high temperature, the two analyses converge, but at cooler temperatures, the HFS of the line enables the spectral line feature to become considerably stronger compared to basing the analysis on a singular atomic level transition. The metallicity and scandium abundance is assumed to be solar, and the surface gravity is assumed to follow isochrone relations with changing temperatures. Non-LTE and 3D effects are not considered.

*2.3. Other Important Physical Effects*

When it comes to accurate abundance analysis, having the correct atomic physics data is not always enough. Other effects need to be considered, like the 3D structure of the stellar atmosphere and its dynamics [26], as well as departure from local thermodynamic equilibrium [27,28]. This work recognizes that these effects are likely to be important for good abundance analysis of scandium in cool stars and therefore encourages this to be investigated further.

## 3. Conclusions

As shown in Figure 1, the strong scandium features found in stellar spectra dd not seem to be connected to the location of the observed star, as otherwise suggested by Do et al. [5]. Rather, it was shown by Thorsbro et al. [6] that an atomic physics approach was needed to explain the formation of strong lines, as shown in Figure 2. In particular, it was shown that the works of Pehlivan et al. [11], van Deelen [12] were needed to understand this line formation, showing that even today, there is still a need to investigate the atomic physics properties of many chemical species. The galactic center therefore did not seem to be that different from the solar neighborhood in this respect.

**Funding:** B.T. acknowledges support from the Swedish Research Council, VR (Project Number 621-2014-5640), Funds from Kungl. Fysiografiska Sällskapet i Lund (Stiftelsen Walter Gyllenbergs fond and Märta och Erik Holmbergs donation).

**Acknowledgments:** B.T. acknowledges the collaboration group through which the published work being reported on here was produced. The collaboration includes Nils Ryde, Michael Rich, Mathias Schulteis, Livia Origlia, Tobias Fritz, Henrik Hartman, Maria Lomaeva, Henrik Jönsson, Hampus Nilsson, Per Jönsson, Asli Pehlivan Rhodin, and Felix van Deelen.

**Conflicts of Interest:** The author declare no conflicts of interest.

## Abbreviations

The following abbreviations are used in this manuscript:

HFS　　Hyperfine structure
LTE　　Local thermodynamic equilibrium

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
