# Peer review of "Atomic Data Needs in Astrophysics: The Galactic Center “Scandium Mystery”"

_atoms, doi:10.3390/atoms8010004_

Round 1
Reviewer 1 Report
A referee report on the manuscript Atoms-6555580 'Atomic data needs in astrophysics: The Galactic centre "scandium mystery" by Brian Thorsbro. The refereed paper is a short version of the recently published large article by Thorsbro et al. ApJ 866, 52, 2018. As I understand it gives an example of how a problem of high scandium abundance in Galactic Centre compared to the solar neighbourhood (found by Do et al. in 2018) may be solved involving the new laboratory data for Sc I IR lines (transition probabilities and hyperfine splitting). As shown in Fig.3 the largest effect in abnormal intensity of Sc I lines is caused by hfs splitting. HFS data are taken from the recent (practically unpublished) work by van Deelen. However, the hfs laboratory data for Sc I are known from the last century and published by Ertmer, W. and Hofer, B. 1976, Z. Phys. A 276, 9-14; Aboussaid, A., Carleer, M., Hurtans, D., Biemont, E., and Godefroid, M.R. 1996. Physica Scripta 53, 28-32. Normally, if the author of the present paper claims the needs for new/improved atomic data it means that the required data either do not exist or they are not good, and in this case the comparison have to be present. I wonder why the author does not even mention the existence of the previous hfs data. It should be done. Other remarks: Abstract, line 9-10. The last sentence sounds strange, please, reformulate it. Also, correct '... can be used to explain'. Introduction: line 29. The statement '..changes little during the life of the star' is not correct. It is correct for Main Sequence life, while the author consider M-giants. Introduction: line 38. The sentence 'More energetic light is scattered away by the dust ...' should be changed by using the common definition 'extinction' which includes both absorption and scattering. Section 2.1, line 63. '...the features of scandium lines diminish as the temperature goes down' ? What does 'the feature of scandium lines' mean ? If it is the intensity, then it increases as the temperature goes down. Figure 1. Please, indicate in the Capture that wavelength scale is in vacuum. Acknowledgements: The paper has only one author, but there are too many persons mentioned in Acknowledgments. Perhaps, this is a tail from the larger paper by Thorsbro et al. (2018).Author Response
Thank you for your feedback, it has improved the quality of the work, and for that I am grateful.
Please find my responses in the attached file.
Best regards,
Brian Thorsbro

Reviewer 2 Report
The paper presents a brief exposition of the Galactic center scandium problem. As shown in the previous work by the authors (Thorsbro et al. 2018), the problem is not due to any peculiar star formation history of the Galactic centre, but rather due to erroneous astrophysical modelling of spectral lines of Sc.
The manuscript is generally appropriate for publication in MDPI, but a few minor issues need to be sorted out, before I can recommend this paper for publication.
(1) The introduction can be compressed by a factor of two. Much of the general and quite simplistic discussion about galaxies, Milky Way, stellar evolution, etc can be removed. The discussion should rather focus on the scope of the present work.
(2) The remainder of available space shall be used to provide more information about the methods proposed in this work, and about the actual results. In the current version of the paper, it is not clear what has been done and what is the main advance over previous work.
Were new measurements of the HFS data obtained? If yes, by whom and how?
Do the authors suggest that the lack of HFS in previous studies (e.g. Do et al. 2018 who employed the SME code) lead to unrealistically over-estimated Sc abundances?
(3) The most relevant graph in the paper is Figure 3 which illustrates the difference between the EWs of Sc lines compued with and without HFS. I think this should be the first figure in the paper.
(4) How about 3D and NLTE effects in Sc? NLTE estimates for Sc lines were discussed in Zhang, Gehren, Zhao 2008; see also Bergemann & Nordlander 2014. 3D estimates are available in the study of Scott, Asplund et al. 2015
Author Response
Thank you for your feedback, it has improved the quality of the work, and for that I am grateful.
Please find my responses in the attached file.
Best regards,
Brian Thorsbro

Reviewer 3 Report
Manuscript “Atomic data needs in astrophysics: The Galactic centre “scandium mystery”
Authors: Brian Thorsbro
Review
This is a well written manuscript that provides a good review about the very interesting topic of the study of the abundance of scandium. Minor comments for improvement are provided.
-Line 59: “Figure 1”, do the authors have permission to use this image?
-Lines 63-64: “scandium lines diminish as the temperature goes down”. Should not it be “up”?
-Line 71: “a program has been initiated”, could a reference be provided?
-Line 80: “using the mixing-length recipe”, could a reference be provided?
-Line 87: “as shown in Fig 2, the strong scandium features DO not seem to be connected to the location of the observed star”.
Author Response

(The authors gave the same response as above.)
